# Macroscopic conductivity of aqueous electrolyte solutions scales with ultrafast microscopic ion motions

Vasileios Balos [1,4], Sho Imoto[1], Roland R. Netz [2], Mischa Bonn[1], Douwe Jan Bonthuis[2,3✉], Yuki Nagata[1✉] & Johannes Hunger [1✉]

Despite the widespread use of aqueous electrolytes as conductors, the molecular mechanism of ionic conductivity at moderate to high electrolyte concentrations remains largely unresolved. Using a combination of dielectric spectroscopy and molecular dynamics simulations, we show that the absorption of electrolytes at ~0.3 THz sensitively reports on the local environment of ions. The magnitude of these high-frequency ionic motions scales linearly with conductivity for a wide range of ions and concentrations. This scaling is rationalized within a harmonic oscillator model based on the potential of mean force extracted from simulations. Our results thus suggest that long-ranged ionic transport is intimately related to the local energy landscape and to the friction for short-ranged ion dynamics: a high macroscopic electrolyte conductivity is thereby shown to be related to large-amplitude motions at a molecular scale.

[1] Department of Molecular Spectroscopy, Max Planck Institute for Polymer Research, Ackermannweg 10, 55128 Mainz, Germany. [2] Fachbereich Physik, Freie Universität Berlin, Arnimallee 14, 14195 Berlin, Germany. [3] Institute of Theoretical and Computational Physics, Graz University of Technology, Petersgasse 16/II, 8010 Graz, Austria. [4] Present address: Department of Physical Chemistry, Fritz Haber Institute of the Max Planck Society, Faradayweg 4-6, 14195 Berlin, Germany. ✉email: bonthuis@tugraz.at; nagata@mpip-mainz.mpg.de; hunger@mpip-mainz.mpg.de

The electrical conductivity of an electrolyte solution is arguably its most critical property, as it limits the electrolyte performance in, e.g. batteries[1], fuel cells[2], or supercapacitors[3]. Yet, the molecular-level understanding of factors improving or suppressing charge mobility is still at its infancy[4]. Consequently, macroscopic measures, such as the product of electrolyte viscosity and single ion conductance (i.e. the so-called Walden product), are common—yet controversial—measures for the electrolyte performance[5]. Even for relatively simple electrolyte solutions, like aqueous salt solutions at moderate concentrations, which are both technologically[6,7] and biologically[8] relevant, it has been challenging to understand long-ranged charge transport on a molecular scale. Our lack of understanding of the microscopic, molecular-level mechanisms determining macroscopic conductivity of electrolyte solutions precludes a rational design of, and search for, new electrolytes.

The challenge in understanding ionic conductance can be traced back to marked structuring[9] and correlated motions[9,10] in electrolytes. An ion cannot diffuse independently without displacement of the surrounding ions[11]. The same restriction applies to ionic solvation shell dynamics:[10] An ion has to strip at least part of its solvation shell to be transported, and exchange of solvent molecules in ionic solvation shells is obviously involved in ion conduction[12]. Thus, molecular-scale motion on a picosecond scale impacts macroscopic charge transport on long timescales (>nanoseconds)[13]. These slow collective dynamics have been theoretically linked to the molecular-level fast (picosecond scale) dynamics for dilute salt solutions[14]. For technologically relevant concentrated electrolytes[1], the fast dynamics of both the ions[15–17] and their solvation shells[10,18–20], have been elucidated using spectroscopic techniques. Yet, the relevance of such fast molecular motion to macroscopic transport has remained elusive[21].

Both, the dynamics of water in the solvation shell of ions and the motion of ions itself, go along with a change of the macroscopic dipole moment of the sample and can thus be probed using spectroscopy experiments[10,22–24]. Here microwave and Terahertz spectroscopies have been extensively used. At field frequencies ranging from 100 MHz to ~100 GHz, hydration of ions has been intensively studied by detecting the dynamics of water and also the rotational dynamics of long-lived ionic aggregates (i.e. ion-pairs) have been elucidated in detail[24–26]. At frequencies ranging from ~2 THz up to 20 THz, at which hydrogen-bonding vibrations, librations, and also ion dynamics contribute, the cooperativity and spatial extent of ion hydration, as well as ion-pairing, have been investigated in great detail[10,23,27,28]. At intermediate frequencies (100 GHz–2 THz) computational studies predict the very weak contribution of ionic currents to peak[22]. Experimentally, this intermediate frequency range is, however, challenging to study and experiments on electrolytes are scarce[29]. As such, the potential of using this spectral information to understand ion dynamics has so far not been exploited[22,29].

Here, by combining GHz to THz dielectric relaxation spectroscopy (DS)[30] and molecular dynamics (MD) simulation, we find that the contribution of ions to these spectra, which peaks at ~0.3 THz, arises from the microscopic "cage" motion of ions in their potential energy minimum. This cage is imposed by (counter)ions and water molecules in their immediate surrounding. Surprisingly, the amplitude of this fast cage motion shows universal scaling with the macroscopic conductivity, independent of the nature of the ions. Using a harmonic potential model, we illustrate that the amplitude of motion in this "cage" is related to the ionic conductivity.

## Results

### Experimental polarization dynamics of aqueous KI solutions.
We probe the response of the electrolytes to an externally applied electrical field with frequency $v$, using broadband DS at frequencies spanning from a few hundred of MHz to ~1.5 THz. DS probes the electrical polarization of a sample in an external field and is thus sensitive to any dynamics that go along with the displacement of charges from their equilibrium positions. Such displacement reflects either the rotation of molecules with an electrical dipole moment or the translation of ions. Typically, the response is expressed as the complex dielectric permittivity, where the response out of phase with respect to the external field is the dielectric loss and representative of absorption of the electric field. The frequency-dependent permittivity is a measure for the polarization in phase with the external field (for details on the experimental determination of permittivity spectra see "Methods" section). For neat water at ambient temperature, DS spectra (Fig. 1a) are dominated by the collective re-orientational motion of the dipolar water molecules at ~20 GHz[31], as evident from the dispersion in the dielectric permittivity and a peak in the dielectric loss. The effect of various electrolytes on the re-orientational motion of water has previously been studied both experimentally[24,26,32] and computationally[22,33,34] in great detail. Adjacent to the dominant relaxation at ~20 GHz, at least one low amplitude mode is present at 0.3–1 THz in neat water (light red shaded area in Fig. 1a)[29,35,36]. The molecular origin of this 0.3 THz dynamics is still under debate[37] and has been ascribed to the relaxation of weakly hydrogen-bonded water molecules[29], to small angular motions of water molecules due to rapid fluctuations[37], or to the dynamics of the low-density liquid phase of water[38].

With increasing concentration of salt (here KI, $c_{KI}$), we find the spectral contributions at ~20 GHz to be reduced. This reduced polarization has been reported for various salts and has been attributed to the reduced correlation between the rotational motion of water molecules in ionic hydration shells[22,34,39] and also—to a lesser extent—to kinetic depolarization[33,40–42]. Conversely, the spectral contributions to the 0.3–1 THz response increase with increasing $c_{KI}$. The inset of Fig. 1a shows an increasing dispersion in the dielectric permittivity and a slight increase in the dielectric loss, as $c_{KI}$ increases: With increasing salt concentration, molecular or ionic motion at these frequencies goes along with an enhanced displacement of charges. To quantify the changes to the spectra, we model the experimental spectra: to describe the main relaxation at ~20 GHz, we use a Cole–Cole mode, in line with previous studies[32]. To capture the faster dynamics at ~0.3 THz, we use a Debye-type mode[32] (see "Methods" section for details). We note that both, motion of ions and motion of water molecules, contribute to the spectra at these frequencies[22]. As such, we use the Debye mode as a means to quantify the contribution of all fast dynamics. The variation of the parameters of the Debye mode with concentration thus reflects the salt-induced changes to these dynamics. Using this approach, we find the relaxation amplitudes for the fast dynamics (relaxation time 200–500 fs) to be very sensitive to the presence of the salt (Fig. 1b): Starting from an initial value of ~2 (neat water) the amplitude increases linearly with increasing concentration of KI, in line with what has been found for solutions of NaCl[29] and several alkali-halide salts at >2 THz[28].

### Dissecting ions' contributions using MD simulations.
To pinpoint the origin of the response at 0.3–1 THz, we perform force-field MD simulations (for details, see "Methods" section, Supplementary Figs. 1–9, and Supplementary Notes 1 and 2). The dielectric response can be readily obtained from the time-correlation function of the macroscopic dipole moment (the electrical dipole moment due to both water molecules and ions). By computing the time-correlation function due to only water

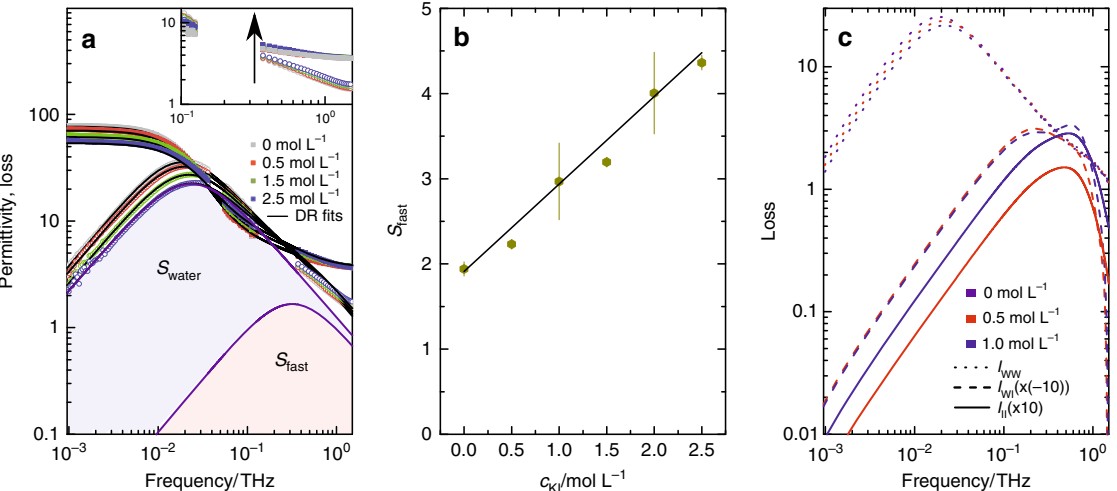

**Fig. 1 Polarization dynamics of solutions of KI from experimental and simulated dielectric spectra. a** Experimental dielectric permittivity (filled symbols) and dielectric loss spectra (open symbols) of aqueous solutions of KI. Lines show fits using the relaxation model (see "Methods" section). The shaded areas show the contribution of the main water mode at 20 GHz (light blue) and the fast mode at ~0.3 THz (light red) to the loss spectrum of the 2.5 mol $L^{-1}$ solution of KI. The 20 GHz mode decreases in amplitude with increasing ionic concentration. In contrast, a zoom-in into the 0.1–1.5 THz spectral range (see inset) reveals increasing spectral amplitudes with increasing KI concentration. Note that for visual clarity, the Ohmic loss (last term of Eq. (1)) has been subtracted. **b** The amplitude of the fast mode as obtained from fits to the experimental spectra as a function of salt concentration exhibits an increase with salt concentration. The solid line shows a linear fit. Error bars correspond to the standard deviation within six independent measurements. **c** Dielectric loss of aqueous solutions of KI at concentrations of 0, 0.5, and 1 mol $L^{-1}$ as obtained from MD simulations (for details see Methods section). Dotted lines show the contribution of only water, solid lines the contribution of only the ions, and the dashed lines refer to the (negative) contribution from the correlation between ionic and water motion (see also Supplementary Fig. 9).

molecules, one can isolate the contribution of water to the spectra ($I_{WW}$ in Fig. 1c). The dominant dielectric loss for water peaks at 20 GHz. The lower peak amplitude in the simulations as compared to the experiments arises from the used TIP4P/2005 water force-field, which is a shortcoming of this force-field[43]. Nevertheless, the TIP4P/2005 model is known to reproduce both structure and (fast) dynamics[44], which are relevant to the present work, very well: the experimentally observed peak shoulder at 0.3 THz is also apparent in the simulated loss spectra.

Upon increasing $c_{KI}$, the polarization dynamics at 20 GHz is reduced, again consistent with the experimental data. In contrast, water's contribution at 0.3 THz is hardly affected by the presence of KI. Hence, the simulations suggest that the experimentally observed salt-induced response at ~0.3 THz does not originate from salt-induced changes to the water dynamics, as opposed to previous interpretations of experimental data;[29] rather, the ions' contribution ($I_{II}$ in Fig. 1c), which is polarization due to the displacement of both anions and cations out of their equilibrium position, increases with increasing salt concentrations. The correlation between the dynamics of water and the ions ($I_{WI}$ in Fig. 1c) has a negative sign and 'counters' the ionic polarization, similar to what has been found at higher frequencies by Marx and coworkers[10]. Nevertheless, the net-polarization due to the presence of salts ($I_{II} + I_{WI}$) is positive at ≳1 THz (for a more detailed discussion on $I_{WI}$, see Supplementary Note 2 and Supplementary Fig. 9). Hence, based on the simulation results, the ion-induced changes to the polarization dynamics at THz frequencies can be primarily ascribed to the (hindered translational) motion of ions.

**Harmonic oscillator to model fast ion motions**. To rationalize the molecular level origins of the observed ion dynamics, we constructed a harmonic potential model. First, we compute the cation–anion radial distribution function (RDF), $g(r)$, for KI. $g(r)$ shows a distinct peak at 3.3–3.8 Å (Fig. 2b), which corresponds to

the first 'shell' of counterions around a given ion. Through the Boltzmann relation $F(r) = -k_B T \ln(g(r))$, we obtain the potential of mean force $F(r)$ for anion–cation pairs (Fig. 2c), where $k_B$ and $T$ are the Boltzmann constant and temperature, respectively[45]. $F(r)$ thus contains excluded volume effects of both water and ions as well as attractive intermolecular interactions. Approximating the potential to be harmonic (for details see "Methods" section), we obtain a force constant of $k = 8.5$ kg $s^{-2}$ for 0.5 mol $L^{-1}$ KI. Together with the reduced mass $\mu = 30$ g $mol^{-1}$ for KI, the frequency of these ion motions is predicted to be $\omega_0/2\pi \approx (k/\mu)^{1/2}/2\pi = 2$ THz (see also Supplementary Note 3). By further assuming the damping coefficient of the ionic motion to be given by the macroscopic hydrodynamic drag coefficient $\gamma = 10^{-12}$ kg $mol^{-1}$ $s^{-1}$ for dilute KI solutions[46], the thus obtained damped harmonic oscillator reproduces the spectral shape and amplitude of the ionic dynamics in the frequency domain very well (purple line in Fig. 2a). The predicted maximum of the spectral response of the harmonic oscillator model is shifted by a factor of ~4 to higher frequencies as compared to the simulated $I_{II}$ response (Fig. 2a), which is attributable to the anharmonicity of the potential and the presence of ions beyond the first coordination shell, neglected in our harmonic approximation (Fig. 2c, see also Supplementary Note 4 and Supplementary Fig. 10). Also an underestimation of the reduced mass, as hydration of ions and/or electrostatic interaction with other ions may effectively result in a higher reduced mass, could contribute to this difference.

Note that while the harmonic oscillator model does not capture all details of the simulated $I_{II}$ spectra, it serves to illustrate the underlying molecular-level dynamics. Within this model, the peak maximum (or similar, the peak integral) of the oscillator inversely scales with damping, as higher friction attenuates the ions' motion and thus reduces the ionic polarization according to $\varepsilon_{HO}''(\omega_0) \sim c_{salt}/\omega_0 \gamma$ (see Supplementary Eq. (13)). The oscillator amplitude also scales with the inverse fluctuation frequency, $\omega_0$, since steeper (shallower) potentials and/or heavier (lighter) ions narrow (widen) the spatial extent of thermally accessible

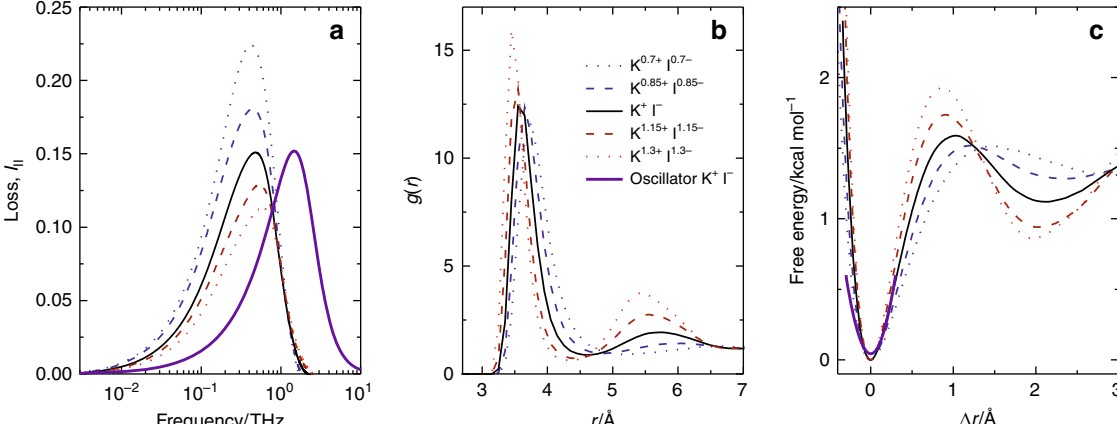

**Fig. 2 Effect of the ionic distribution on the ionic dielectric response from charge-scaled MD simulations. a** Ionic contribution $I_{II}$ to the dielectric loss as obtained from MD simulation of 0.5 mol L$^{-1}$ KI (solid black lines). To explore the effect of the ionic distribution, we also show the contribution based on the distributions obtained for KI with increased (1.15 and 1.30) and decreased (0.7 and 0.85) ionic charge (see legend in panel **b**). Note that charge scaling was used to alter the distribution of ions. To isolate the effect of the distribution, we calculate $I_{II}$ assuming ions with unity charges. The purple curve in panel **a** shows the harmonic oscillator model for unity charged ions. **b** The altered ionic charge results in marked differences in the cation–anion RDF $g(r)$: Increasing ionic charge results in sharper (more structured) peaks. **c** The cation–anion potential energy surface inferred from the RDFs. With decreasing charge density, the ions reside in shallower potentials. The purple line shows the harmonic approximation for the unity charged ions, which gives the purple spectrum shown in panel **a**.

excursions of the ions out from their equilibrium. We conclude that the $I_{II}$ spectral contribution sensitively reports on the ionic distribution and dissipative ion effects (as quantified by the friction coefficient) in solution.

**Effect of nature of the ions on fast motion.** The sensitivity of the ions' response to the ionic distribution can be directly demonstrated using MD simulations. To this end, we reduce (or increase) the charge of the ions. Altered ionic charges lead to altered interaction with water, leading to a modification of the friction experienced by the ion and a marked change in the width of the first ionic coordination peak in the RDFs (Fig. 2b). As such, the potential of mean force of the ions becomes more shallow (steep), and based on the oscillator model, the magnitude of the fluctuations is expected to be accordingly enhanced (reduced). Quantitatively, we find the height of the simulated $I_{II}$ peaks (Fig. 2a) to vary by +50% (−25%), upon decreasing (increasing) the charge by 30%, while the diffusivities of the ions vary by +30% (−30%) (see Supplementary Table 1). In light of the oscillator model described above, the higher sensitivity of $I_{II}$ to the ionic charge, as compared to the diffusivity, shows that both the altered friction and the altered ionic distribution give rise to the changes in $I_{II}$ upon charge scaling. Thus, the ionic contributions to the spectra sensitively report on the interaction and distribution of ions in electrolytes.

To demonstrate the sensitivity of the ~0.3 THz absorption to the ionic species experimentally, we measure the DS spectra for different ions at various concentrations (for spectra and fits see Supplementary Figs. 11–17). As the nature of the ions and their concentration markedly affects their pair distribution, their dynamics are expected to be ion-specific, and the magnitude of the ionic polarization will depend on both concentration of electrolyte and the nature of the salts (e.g., ionic radii, van der Waals repulsion, "soft or hard" nature of the ions, etc.). In contrast to the simulations, which can disentangle the spectral contribution of ions from the contribution of water, we experimentally monitor all fast dynamics due to water and ions by studying the amplitude of the fitted Debye mode. As the motions of both water and ions contribute to the spectral

intensity at ~0.3 THz (Fig. 1c), only the variation of the intensity with salt concentration allows for drawing conclusions on ion-induced dynamics, while the absolute values of $S_{fast}$ (and also $\tau_{fast}$) also contain information on the dynamics of water. As can be seen in Fig. 3a, the values for $S_{fast}$ vary widely for different mono- and bivalent salts. However, salt concentration does not exclusively determine the magnitude of $S_{fast}$ when comparing all studied salts (as would follow from asymptotic electrolyte theories that treat ions as point-like charges): The data in Fig. 3a are scattered, and we find a Pearson's correlation coefficient $r = 0.64$ for $S_{fast}$ ($c_{salt}$).

Conversely, the oscillator model suggests that the amplitude of the ionic contributions scales with the inverse damping and the ions concentration. Thus, given that the macroscopic friction for long-range transport and the friction for the short-scale THz motion are correlated, the spectral contributions of the ions to the fast dynamics are expected to scale with the macroscopic conductivity (see "Methods" section), rather than with concentration (see also Supplementary Note 5 and Supplementary Fig. 18). Moreover, the curvature of the potential for the ions, which also affects the spectral contributions, may also be related to the energetic barrier to escape the potential minimum (for our simulations on solutions of KI the height of the maximum at $\Delta r \approx 1$ Å in Fig. 2c is related to the curvature at the minimum, see Supplementary Note 6 and Supplementary Fig. 19). This suggests, that the ionic dynamics at ~0.3 THz are related to the energy barrier for the ion to 'escape' the solvation cage: The shallower the cage potential, the easier the ion can escape its solvation cage. Translating the ion out of the solvation cage is again the ionic conductivity.

To experimentally testify the relation between local ion dynamics and macroscopic conductivity, we plot in Fig. 3c the experimentally obtained amplitude of the fast dynamics $S_{fast}$ as obtained from fitting the relaxation model to the experimental spectra vs. the electrolyte conductivity $\kappa$. Remarkably, in contrast to the moderate correlation of $S_{fast}$ with $c_{salt}$, the values of $S_{fast}$ show a very strong correlation with the electrolyte conductivity (Pearson's correlation coefficient $r = 0.82$, Fig. 3c) for a wide range of monovalent and bivalent salts (CsCl, KCl, NaCl, LiCl, GdmCl, KI, KSCN, Na$_2$SO$_4$, MgSO$_4$, MgCl$_2$): Starting from a

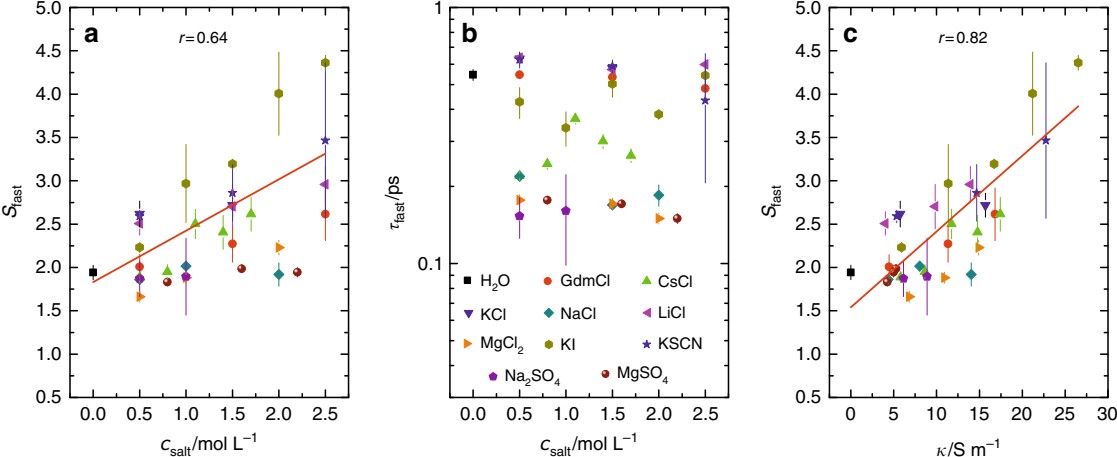

**Fig. 3 Effect of the ion type on the spectroscopically observed fast dynamics. a** Amplitude, $S_{fast}$, of the fast mode as a function of salt concentration, $c_{salt}$. The variation of $S_{fast}$ with $c_{salt}$ largely varies with the salt type. The solid line shows a linear fit to all data, with a Pearson's correlation coefficient of $r = 0.64$. **b** Relaxation time of the fast Debye mode as a function of salt concentration. **c** $S_{fast}$ as a function of electrolyte conductivity, $\kappa$. The solid lines show a linear fit to all data, with a Pearson's correlations coefficient of $r = 0.82$, demonstrating the scaling of $S_{fast}$ with conductivity. Error bars in all panels correspond to the standard deviation within six independent measurements.

value of ~2 for the fast dynamics at ~0 S m$^{-1}$ (neat water), the amplitude increases to ~3.5 for electrolytes with a d.c. conductivity of ~25 S m$^{-1}$. We note that even for the strong acid HCl the correlation holds at low concentrations, while it breaks down at higher concentrations (see Supplementary Fig. 20), which can be related to the very different charge transport mechanism for the proton (Grothuss-type transport with charge transport being decoupled from mass transport[47]). The experimentally determined Debye relaxation times (~inverse center frequency) of the fast relaxation lie in the range 100–700 fs (Fig. 3b).

**Spatial correlation of ions' motion.** Together, the experimental and simulation results show that short-ranged motions of ions and long-ranged diffusive transport of ions are correlated (for quantitative comparison with the harmonic oscillator model, see Supplementary Note 7 and Supplementary Fig. 21), even at relatively high salt concentrations. This correlation stems in part from similar trends in friction. For concentrated electrolytes, friction governing conductivity contains not only hydrodynamic (Stokes friction) contributions, but also ionic friction (ion cloud relaxation and electrophoretic drag, see also Supplementary Fig. 14a)[11]. Thus, ionic motions are highly correlated, and one question that remains is the degree of collectivity of the ionic motion at ~0.3 THz discussed here. The simple harmonic oscillator model based on the RDF takes only ion pairs into account. The dynamics in the first coordination shell are not independent of the motion of ions in the other coordination shells, an effect that is not contained in the RDF. To determine the degree of collectivity of the ionic dynamics, we performed MD simulations with water molecules beyond the first, second, and third hydration shell fixed in their coordinates. These simulations for a 0.5 mol L$^{-1}$ KI solution (Fig. 4) indicate that the ionic dynamics approach the bulk dynamics only if molecules within the third coordination shell are mobile. This means that the observed ion dynamics at THz frequencies for the 0.5 mol L$^{-1}$ KI solution (Fig. 4) are governed by the correlated motions of ions and water, with the correlation extending up to three coordination shells (~0.9 nm). This correlation length is comparable to the ~0.4 nm Debye length for a 0.5 mol L$^{-1}$ salt solution. Hence, despite classical mean-field theories like Debye–Hückel being unable to predict the observed ion dynamics as they cannot explicitly

account for the interaction between ions and water, the Debye screening length still provides a satisfactorily accurate estimate for the spatial extent of ionic correlations.

## Discussion

In summary, we show that the fast (~0.3 THz) dynamics in the dielectric relaxation spectra of electrolytes reflects the dynamics of ions in their solvation cages. These dynamics are ion-specific, but the magnitude of the motion scales universally with conductivity. This means that the microscopic motion of the ions in a given solvation cage and the macroscopic transport of the ions are intimately connected. This correlation can be understood by noting that microscopic and macroscopic frictions are related and that the potential energy landscape of the ions in the cage determines the ions' thermally accessible excursions, which are also related to the ion escaping its cage to allow for ion transport. This includes the frequently inferred concept of ion-pairing[23,48]: electroneutral ion-pairs do not contribute to conductivity, and the ions reside in very steep potentials that restrict the amplitude of the ionic motions. Although a simple harmonic oscillator model for the ions' motion captures large parts of the observed polarization dynamics, correlated motion involving coordination shells beyond the classical Debye length has to be included to fully capture the observed polarization dynamics. Despite the Debye length being a reasonable estimate for the spatial extent, the ionic dynamics, in fact, probe ionic distributions that go beyond a classical Debye–Hückel charge distribution, as Debye–Hückel does not take the molecular nature of the solvent (cage potentials) into account. As such, the dynamics of ions at 0.1–1 THz are a sensitive experimental measure for the ion distribution in concentrated electrolyte solutions that goes beyond mean-field theories and provide a means to test more advanced electrolyte theories that can explicitly account for molecular solvents. The scaling reported here provides a rationale for understanding, and possibly engineering, the macroscopic conductivity in electrolytes, e.g., ionic liquids or battery electrolytes.

## Methods

**Samples**. CsCl, NaCl, MgCl$_2$·6H$_2$O, KCl, KI, KSCN, LiCl, GdmCl, Na$_2$SO$_4$, and HCl were purchased from Sigma Aldrich, and MgSO$_4$ was purchased from Carl Roth. Aqueous salt solutions of KI (0.5, 1.0, 1.5, 2.0, and 2.5 mol L$^{-1}$), CsCl (0.5, 0.8, 1.1, 1.4, and 1.7 mol L$^{-1}$), KSCN, LiCl, GdmCl (0.5, 1.5, and 2.5 mol L$^{-1}$), NaCl, MgCl$_2$ (0.5, 1, and 2 mol L$^{-1}$), Na$_2$SO$_4$ (0.5 and 1 mol L$^{-1}$), MgSO$_4$ (0.8,

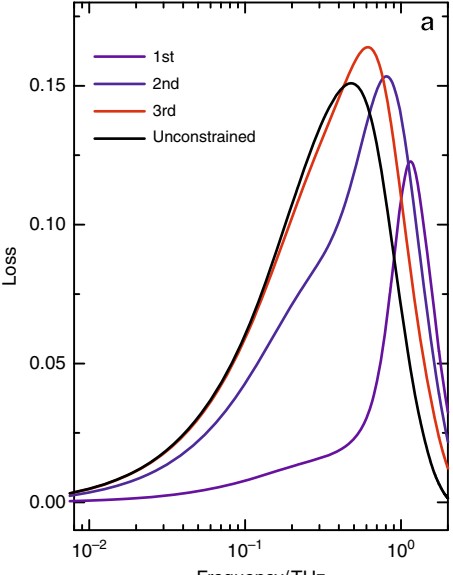
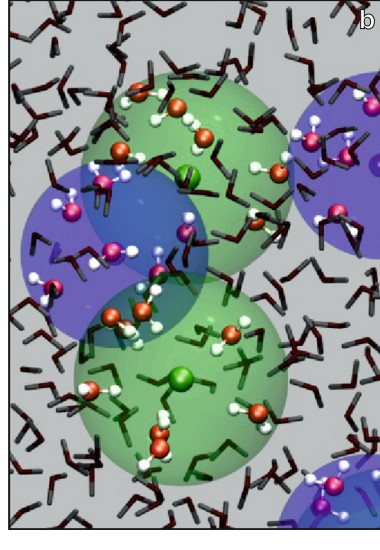

**Fig. 4 Spatial extent of fast ionic motions from MD simulations. a** Ionic contributions ($I_{\parallel}$) to the dielectric loss spectra as obtained from molecular dynamics simulations of 0.5 mol L$^{-1}$ KI. To explore the spatial extent of the correlation of the ionic motion, we fix the coordinates of water molecules beyond the first (purple line), second (blue line), and third (red line) hydration shell. The spectra computed with mobile molecules up to the third hydration shell of the ions (red line) resembles the spectral contribution for unconstrained motion (solid black line). **b** MD snapshot illustrating the constrained simulations. Coordinates of water molecules within the shaded spheres are mobile, while other molecules are fixed in their coordinates. The mobile water molecules are shown as ball-and-stick models.

1.6 and 2.2 mol L$^{-1}$), KCl (0.5 and 1.5 mol L$^{-1}$) and HCl (0.3, 0.4, 0.5, 1, and 2 mol L$^{-1}$) were prepared by weighing the salts into volumetric flasks (for hygroscopic salts in a glove box) and subsequently filled with Milli-Q water.

**Dielectric relaxation spectroscopy.** Complex permittivity spectra, $\hat{\varepsilon}(\nu) = \varepsilon'(\nu) - i\varepsilon''(\nu)$, were recorded at $0.96 \leq \nu/\text{GHz} \leq 125$ using a frequency domain reflectometer based on an Anritsu Vector Star MS4647A[40,49]. Terahertz frequencies ($0.3 \leq \nu/\text{THz} \leq 1.5$) were covered with a Terahertz time domain spectrometer[49] with the samples contained in a fused silica cuvette with a path length of 100 μm[50]. All experiments were performed at 296 ± 2 K.

In line with literature reports[29,31,35,40,51], we fit a relaxation model based on a Cole–Cole equation and a Debye mode to the spectra:

$$\hat{\varepsilon}(\nu) = \frac{S_{\text{water}}}{1 + (2\pi i \nu \tau_{\text{water}})^{(1-\alpha_{\text{CC}})}} + \frac{S_{\text{fast}}}{1 + 2\pi i \nu \tau_{\text{fast}}} + \varepsilon_\infty + \frac{\kappa}{2\pi i \nu \varepsilon_o} \quad (1)$$

The first (Cole–Cole) term of Eq. (1) represents the main relaxation at ~20 GHz with relaxation time, $\tau_{\text{water}}$, the relaxation strength, $S_{\text{water}}$, and $\alpha_{\text{CC}}$ the Cole–Cole parameter[52]. The second (Debye) term models the fast mode at ~0.3 THz with relaxation time, $\tau_{\text{fast}}$, and relaxation strength, $S_{\text{fast}}$. The limiting permittivity $\varepsilon_\infty$ subsumes all polarization components at higher frequencies. The last term of Eq. (1) accounts for Ohmic loss contributions due to the electrolyte conductivity, with $\varepsilon_0$ the permittivity of free space. We assume the conductivity, $\kappa$, to be real and independent of frequency. Thus, any frequency-dependent conductivity will be modeled by the Cole–Cole and the Debye term. Equation (1) was used to model the spectra for solutions of HCl, KCl, KI, LiCl, CsCl, NaCl, MgCl$_2$, and GdmCl. Obtained parameters are shown in Fig. 3 and Supplementary Figs. 13 and 14. For solutions of KSCN, Na$_2$SO$_4$, and MgSO$_4$, an additional Debye term was used to describe rotational motions centered below 2 GHz of the non-centrosymmetric SCN$^-$[40], NaSO$_4^-$ ion-pairs[53], and MgSO$_4$ ion-pairs[54], respectively (for details see Supplementary Note 8 and Supplementary Figs. 15–17). Spectra were fit reducing the deviations of both $\varepsilon'(\nu)$ and $\varepsilon''(\nu)$ on a logarithmic scale (for fits using linear deviations see Supplementary Note 9 and Supplementary Figs. 22 and 23).

**Molecular dynamics simulations.** The MD simulations were performed using the CP2k package[55]. We used the TIP4P/2005 model[43] for water. The potentials for K$^+$ and I$^-$ ions are taken from ref. [56]. The long-range part of the electrostatic interactions was computed using the particle mesh Ewald scheme. The 0 mol L$^{-1}$ system contained 1600 water molecules. The 0.5 and 1.0 mol L$^{-1}$ KI solutions consisted of 14 and 29 KI ion pairs together with 1572 and 1542 water molecules, respectively. The neat water, 0.5 mol L$^{-1}$ KI, and 1.0 mol L$^{-1}$ KI systems were contained in cubic periodic cells with lengths of 36.343, 36.266, and 36.246 Å, respectively. To vary ionic distributions, we scaled ions' charges from ±0.7e to ±1.3e. All other parameters were kept identical to the 0.5 mol L$^{-1}$ KI system. All

systems were equilibrated for 180 ps in the canonical (*NVT*) ensemble using the CSVR thermostat set to 300 K[57]. After equilibration, 40 initial conditions were sampled from the canonical simulations at time intervals of 200 ps to initialize 2 ns long microcanonical (*NVE*) simulations. We also performed microcanonical simulations with fixed molecules (fixed-*NVE*) at 0.5 mol L$^{-1}$ KI. The initial configurations of the fixed-NVE simulations were identical to the *NVE* simulations, whereas the position of ions and water molecules beyond the cut-off radii ($r^c_{K^+}$ and $r^c_{I^-}$) were constrained in the Cartesian space. The cut-off radii were determined from the first, second, and third minimum of the ion-oxygen RDFs ($r^c_{K^+}$, $r^c_{I^-}$)/Å = (3.6, 4.1), (5.8, 6.4), and (8.0, 9.0), respectively (see Supplementary Fig. 2). A time step of 2 fs was used for all simulations and trajectories were saved every 0.2 ps. Radial distribution functions are shown in Supplementary Figs. 1 and 2 and diffusion coefficients obtained from these simulations are given in Supplementary Figs. 3–5 and Supplementary Table 1. The dielectric spectra were calculated through Fourier transformation of the system polarization (for details see Supplementary Note 2, Supplementary Figs. 7–9).

**Harmonic oscillator model.** We approximate the contributions of the ions to the dielectric response by a harmonic oscillator:

$$\hat{\varepsilon}_{HO}(\omega) = \frac{2c_{\text{salt}}q^2}{\varepsilon_0(k - i\omega\gamma - \mu\omega^2)} \quad (2)$$

where $c_{\text{salt}}$ is the salt concentration, $q$ the ions charge, $\omega = 2\pi\nu$ the angular frequency, and $\mu$ the reduced mass. The force constant, $k$, was obtained from fitting a harmonic potential ($F(\Delta r) = 1/2k\Delta r^2$) to the potential of mean force (Fig. 2c). The drag coefficient $\gamma$ ($=k_B T/D$) was approximated based on the experimental diffusivity $D = 0.4$ Å$^2$ ps$^{-1}$[46]. Expressing $\gamma$ in terms of the conductivity $\gamma = c_{\text{salt}}q^2/\kappa$, yields the maximum in the dielectric loss at $\omega_0 = \tau^{-1} \approx \sqrt{k/\mu}$ to scale with conductivity:

$$\varepsilon_{HO}''(\omega_0) \approx \frac{2\kappa\tau}{\varepsilon_0} \quad (3)$$

For further details, see Supplementary Note 3.

## Data availability
Data supporting the findings of this study are available within the article and its Supplementary Information, or from the corresponding authors upon reasonable request.

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

## Acknowledgements

We thank Dr. Bogdan Marekha for fruitful discussions and Martina Knecht for support in recording the dielectric spectra. This work was funded by the Deutsche Forschungsgemeinschaft (DFG) through grants HU1860/4 and SFB 1078 and by the European Research Council (ERC) under the European Union's Horizon 2020 research and innovation program (grant agreement n°714691). JH, RN, and MB acknowledge funding from the Max Planck Society within the "MaxWater" program.

## Author contributions

J.H. conceived and supervised the project. V.B. and J.H. recorded and analyzed the dielectric spectra. S.I. and Y.N. performed the molecular dynamics simulations. S.I., R.N., D.J.B., and Y.N. analyzed the simulation results. D.J.B and R.N. established the harmonic oscillator model. V.B., S.I., R.N., D.J.B., M.B., Y.N., and J.H. discussed and interpreted the results. V.B., D.J.B., M.B., Y.N., and J.H. wrote the manuscript.

## Competing interests

The authors declare no competing interests.
