## [Peer Review File · Nature Communications]

Reviewers' comments:

Reviewer #1 (Remarks to the Author):

Macroscopic conductivity of aqueous electrolyte solutions scales with ultrafast microscopic ion fluctuations

Vasileios Balos, Sho Imoto, Roland R. Netz, Mischa Bonn, Douwe Jan Bonthuis, Yuki Nagata, and Johannes Hunger

The authors present in this paper dielectric spectra up to 1.5 THz. They fit these spectra to a Cole-Cole equation in combination with a Debye mode. The first term represents the well-known water relaxation mode at 20 GHz while the second term, the Debye mode, is fit to a simple model of a harmonic oscillator based on a potential of mean force. They propose, that the amplitude of the latter term, centered at 0.3 THz, is correlated with the conductivity. The authors postulate that this amplitude is related to the energy barrier for ionic transport and conclude that the macroscopic conductivity is related to large amplitude fluctuations at a molecular scale.

Furthermore, they present a harmonic oscillator model based on the potential of mean force and claim that in general the energetic barrier for charge transport can be traced back to this potential barrier for short range fluctuations.

1. While some of their ideas are interesting, I think that at present a publication yields only a minor step beyond the current knowledge as stated in the literature.

The underlying idea that the dielectric spectrum of solvated ions can be traced back to a Cole-Cole distribution representing the solvent water along with a higher frequency contribution from an ion-complex structure is a well known concept. It is schematically shown in Figure 22 of the review from Kaatze of dielectric spectroscopy (*J. Solution Chemistry*, 26, 1049 (1997)).

Buchner et al. have investigated previously, and they reported on dielectric spectroscopy of ion-pairing and hydration in the frequency range up to ca. 100 GHz. They were able to assign three dispersion steps, which can be assigned—with falling relaxation time—to the tumbling motion of ion-pairs, the co-operative relaxation of the H-bond network of 'bulk' water, and the fast reorientation of mobile H₂O molecules. *Phys. Chem. Chem. Phys.*, 2002,4, 2169-2179 or *J. Phys. Chem. B* (2007) 11130, 9010-9017

A recent review on "Ion hydration and Ion Pairing as probed by THz Spectroscopy" summarizing the THz response for 38 salt solutions has been published by Schwaab, Sebastiani and Havenith *Angewandte Chemie* 58, 3000 (2019). These spectra were dissected into an ion specific response and an ion unspecific response, which was assigned to a correlated mode of hydration water. The absorption was assigned to rattling modes for strongly hydrating ions (separately for the cation or anion) and vibrationally induced charge fluctuations in the case of weakly hydrating ions and allowed quantification of ion pairing.

In a previous paper by the Netz group in *J. Chem. Phys.* 141, 214502 (2014) a correlation between random ion fluctuations and conductance was already stated:

"In our simulations, the spectral contribution $\chi''(f)$ shows a red-shift as the anion size increases, which simply reflects that larger ions move more slowly through water. Interestingly, the static dielectric contribution from ion positional fluctuations $\chi''(f=0)$ scales roughly linearly with the salt concentration for NaCl, which indicates that it is due to uncorrelated and random ion positional fluctuations. We see, not surprisingly, that free ions FI exhibit the largest conductance, DSIP, and SIP show a reduced conductance, and CIP ions have a conductance that is typically reduced by more than a factor of two. Note that ion pairs contribute to the conductance by rotation and by small changes in the ion separation."

2. The experimental data (as shown in Figure 3) do not support the strong claims being made by the

authors. The data would require an improved analysis and much more data points in order to come to unambiguous conclusions.

In the following I want to address this point in more detail:

First I want to point out that the density of the solution is changing when increasing the concentration. The experimental data should be density corrected before any fit is done. I cannot find any statement on this in the paper.

A summary of the experimental data is given in Figure 3a): Relaxation amplitude, S_{fast} , of the fast relaxation mode as a function of salt concentration, c_{salt} is displayed. They show linear fits to the data. For a concentration of 0, i.e. bulk water, a value of ca. 2.2 was fixed. What does this offset correspond to? The overall variation in S_{fast} (from 1.5 to 4.0) is rather small compared to this offset value.

Now, I want to take a closer look at the data:

A linear scaling with salt concentration is found for LiCl, KCl, KI, and NaSO₄ (only 2 points).

For GdmCl (3 points), MgSO₄ (3 points), KSCN (2 points only !) the displayed fit to a linear function does not make any sense. The points are far off the fitted line, they have large error bars, and do not show a linear trend.

In the previous studies cited above it was shown that the response is linear for ions like KCl, but becomes nonlinear in case of ion hydration. A nonlinear concentration dependence can be attributed directly to ion pairing.

The authors state: "Remarkably, for a wide range of salts (KCl, LiCl, GdmCl, KI, KSCN, Na₂SO₄, MgSO₄) the experimentally determined amplitudes collapse onto a single curve: Starting from a value of ~2.1 for the fast dynamics of water at ~0 S m⁻¹ (neat water), the amplitude increases to 3.5 for electrolytes with a d.c. conductivity of ~25 S m⁻¹. Even for the strong acid HCl, where the charge transport mechanism is very different (Grothuss type transport with charge transport being decoupled from mass transport⁴⁶), the correlation between the conductivity and the fast relaxation strength holds."

Fig 3 c is not supporting this strong statement. Instead, it shows that the conductivity scales linearly with concentration for only a few ions (LiCl, KCl, and KI), which also do not show ion pairing. However, this result is not unexpected, the conductivity is expected to scale with concentration.

It can be seen that for the same value of $S_{fast} = 2.5$, conductivity values between ca. 3.5 and 12 can be found. Thus the statement in the paper as given above is not supported by the data, and these data do not collapse into a simple linear relationship!

Instead I think the data can be summarized alternatively in the following way: Higher S_{fast} values are found for all salts which contain K and which do not form ion pairs. Lower values are found for ion pair forming salts.

The authors state that "The relaxation time is rather similar for all studied salts, except for sulfate salts, which are prone to the formation of long-lived ion-pairs". The same result was reported before in the review in *Angewandte Chemie* "The linewidths for all these bands reflect the damping of the vibrational modes of the ion-water complexes. Remarkably, we found that all free ions seem to be subject to identical damping, which means that they are all connected to the same thermal bath"

2. The authors fit the data using a damped harmonic oscillator, using the model of Schmidt et al. reference 38 for the frequency range between 50 and 300 cm⁻¹.

However, especially in the low frequency regime, a refined model should be used to fit the overdamped modes, following the Approach described in the book:

A. Nitzan, *Chemical Dynamics in Condensed Phases: Relaxation, Transfer, Reactions in Condensed Molecular Systems*, Oxford University Press, Oxford 2006.

Especially the case of HCl, where charge transport occurs via a Grotthus mechanism, shows that the underlying model is much too simplified.

4. In Figure 4, they display the ionic contributions of the dielectric loss spectrum of a simplified electrolyte showing that the spectra show contributions up to the third shell.

Previously, using ab initio simulations, Schienbein et al have showed that the theoretical THz difference spectra of aqueous salt solutions can be deciphered in terms of only a handful of dipolar auto- and crosscorrelations, including the second solvation shell, see J. Phys. Chem. Lett. 2017, 8, 2373–2380. This very detailed analysis showed that “Dramatic intensity cancellations due to large positive and negative contributions are found to effectively shift intensity maxima. “. For Cl⁻ the largest contribution below 50 cm⁻¹ arises from $\Delta C_{Ion}(\omega)$, which stems from the autocorrelation of the ion and its cross-correlations with water molecules not only in the first, but also in the second solvation shell. However, other negative contributions would become relevant for Br⁻.

Minor point: The last sentence “points towards the possibility of tuning the macroscopic conductivity in e.g. ionic liquids or battery electrolytes by molecular engineering “ does not make sense, since it is not related to any aspect of the paper

Reviewer #2 (Remarks to the Author):

This is a study of the molecular dynamics in the GHz-THz region of aqueous electrolyte solutions by dielectric relaxation spectroscopy and MD simulations. Currently, aqueous solutions of high concentrated electrolytes are getting more attention for application of battery, in particular. In this study, the authors have experimentally found a mild correlation between the electrical conductivity and the amplitude of the fast band at ~ 0.3 THz. MD simulations have pointed out that the THz fluctuations observed in the experiments are governed by the correlated motions of ions and water with the correlation extending up to the 3rd coordination shell which is longer than the Debye length. I think that the findings are interesting and helpful for the deeper understanding of aqueous electrolyte solutions. The manuscript is well-written overall. Though I have some (relatively minor) comments shown in below, I recommend this manuscript for publication in Nature Communications after they are addressed appropriately.

(1) Relation between the electrical conductivity and the amplitude of the 0.3 THz band. I think that this new finding is very interesting. On the other hand, I do not understand it well. My concern is why the “amplitude” (not “time” or “frequency”) of the band is related to the electrical conductivity? The amplitude of the band in the dielectric relaxation is coming from the concentration of the signal origin and the strength of the transition dipole moment. In the case of a single solute, the latter effect should be minor unless ions make some aggregations. However, if the solute changes like this study, the latter effect should be large. It is rather straight forward to understand that the concentration of solute is related to the electrical conductivity, but I do not understand how the strength of the transition dipole moment can influence the electrical conductivity. Adding some comments on this in the text would be helpful to understand the relation.

(2) Terminology of “relaxation”. In this manuscript, the authors use “fast relaxation” for the band at ~0.3 THz (for example, page 8), which is attributed the ions’ fluctuation. It is commonly used the word “relaxation” for an overdamped motion. But I feel that it might be not very appropriate for using “relaxation” for “fluctuation”, an “underdamped motion”, or “vibrational motion”. When it is used for a “vibrational motion”, it often means its dephasing process. If the authors assign the motion at ca. 0.3 THz as the ions’ fluctuation, I recommend to use other word, such as motion or dynamics.

(3) Harmonic oscillator model. It is a bit surprising and not surprising as well that the harmonic

oscillator model can work for aqueous KI solution. The authors simply consider the pair of K^+ and I^- . If I look at Figure 4, however, the situation is not like that simple even in 0.5 M: it seems to be necessary to consider the many body effect. In such case, is the reduced mass appropriate? Some comments for this in the text should be added.

(4) Figure 3. It is hard to distinguish between KCl and $MgSO_4$. I recommend replacing with a distinguishable color for either one.

(5) Typo. Page 8 line 8: "3 coordination shell" should be "3rd coordination shell".

Reviewer 1

Macroscopic conductivity of aqueous electrolyte solutions scales with ultrafast microscopic ion fluctuations

Vasileios Balos, Sho Imoto, Roland R. Netz, Mischa Bonn, Douwe Jan Bonthuis, Yuki Nagata, and Johannes Hunger

The authors present in this paper dielectric spectra up to 1.5 THz. They fit these spectra to a Cole-Cole equation in combination with a Debye mode. The first term represents the well-known water relaxation mode at 20 GHz while the second term, the Debye mode, is fit to a simple model of a harmonic oscillator based on a potential of mean force. They propose, that the amplitude of the latter term, centered at 0.3 THz, is correlated with the conductivity. The authors postulate that this amplitude is related to the energy barrier for ionic transport and conclude that the macroscopic conductivity is related to large amplitude fluctuations at a molecular scale.

Furthermore, they present a harmonic oscillator model based on the potential of mean force and claim that in general the energetic barrier for charge transport can be traced back to this potential barrier for short range fluctuations.

1. While some of their ideas are interesting, I think that at present a publication yields only a minor step beyond the current knowledge as stated in the literature.

We thank the reviewer for finding our idea interesting. As opposed to various literature reports, which focused either on water and ion(-pair) dynamics observed below 100 GHz or above 2 THz, our study reports on the intermediate spectral range (300 GHz – 1 THz), which we find to capture both fast water dynamics and ionic dynamics. The dynamics of aqueous electrolytes in this frequency range has so far been not elucidated. Here, we find that the increase in spectral contributions can be ascribed to ion dynamics.

The combined revisions to the manuscript as a response to the reviewer's first point are given below.

The underlying idea that the dielectric spectrum of solvated ions can be traced back to a Cole-Cole distribution representing the solvent water along with a higher frequency contribution from an ion-complex structure is a well known concept. It is schematically shown in Figure 22 of the review from Kaatz of dielectric spectroscopy (*J. Solution Chemistry*, 26, 1049 (1997)).

Buchner et al. have investigated previously, and they reported on dielectric spectroscopy of ion-pairing and hydration in the frequency range up to ca. 100 GHz. They were able to assign three dispersion steps, which can be assigned—with falling relaxation time—to the

tumbling motion of ion-pairs, the co-operative relaxation of the H-bond network of 'bulk' water, and the fast reorientation of mobile H₂O molecules. Phys. Chem. Chem. Phys., 2002,4, 2169-2179 or J. Phys. Chem. B (2007) 11130, 9010-9017

We agree with the reviewer that in these earlier papers by Kaatze or Buchner (and also others), which were limited to frequencies <89 GHz, the experimental spectra have been described with a higher frequency Debye mode (as in our study) and lower frequency mode(s) to model the experimental spectra. However, discussion of this higher frequency Debye mode is omitted in these papers, as it is centred outside their experimental frequency range. To the best of our knowledge, the only rigorous attempt to study this spectral range experimentally for aqueous salt solutions has been reported by Vinh et al (J. Chem. Phys. 142, 164502 (2015), ref. 29). While this pioneering work on solutions of NaCl finds increasing spectral contribution with increasing salt concentrations, consistent with our findings, the origin of the increase remained elusive. In particular, the contribution of ions, which we show to dominate the salt-induced changes to the fast dynamics, was not considered. In our manuscript, we unveil these contributions and provide their molecular-level origins using simulations, theory, and experiment.

A recent review on "Ion hydration and Ion Pairing as probed by THz Spectroscopy" summarizing the THz response for 38 salt solutions has been published by Schwaab, Sebastiani and Havenith *Angewandte Chemie* 58, 3000 (2019). These spectra were dissected into an ion specific response and an ion unspecific response, which was assigned to a correlated mode of hydration water. The absorption was assigned to rattling modes for strongly hydrating ions (separately for the cation or anion) and vibrationally induced charge fluctuations in the case of weakly hydrating ions and allowed quantification of ion pairing.

We agree that dynamics at higher frequencies >2 THz have been studied in great detail by Havenith and coworkers and, besides hydrogen-bonding vibrations and librations of water, also ions contribute to this spectral range. Yet, the full frequency range of our present work is not covered by these very comprehensive and insightful studies and, in particular, the dynamics of ions at frequencies relevant to our work could not yet be resolved in these studies (see e.g. Fig. 4 of *Angewandte Chemie* 58, 3000 (2019), ref. 23). Thus, our work focuses on a different aspect of dynamics of aqueous electrolytes, reflected in a different frequency range.

In a previous paper by the Netz group in *J. Chem. Phys.* 141, 214502 (2014) a correlation between random ion fluctuations and conductance was already stated:

"In our simulations, the spectral contribution $\chi I(f)$ shows a red-shift as the anion size increases, which simply reflects that larger ions move more slowly through water. Interestingly, the static

dielectric contribution from ion positional fluctuations $\chi_I(f=0)$ scales roughly linearly with the salt concentration for NaCl, which indicates that it is due to uncorrelated and random ion positional fluctuations. We see, not surprisingly, that free ions FI exhibit the largest conductance, DSIP, and SIP show a reduced conductance, and CIP ions have a conductance that is typically reduced by more than a factor of two. Note that ion pairs contribute to the conductance by rotation and by small changes in the ion separation.”

We find this quote from the reviewer somewhat problematic, as different parts of it originate from different locations in the cited manuscript. Most of the quoted text refers to discussion of the d.c. conductivities obtained from simulations. Free ions have naturally a larger contribution to the d.c. conductivity of than paired ions.

Only the first sentence of the reviewer’s ‘quote’ refers to the ionic motion as reported in here. With respect to the variation of these contributions with ionic nature, i.e. the focus of the present work, some of us already back then emphasized that *“no clear and systematic ion-specific trends can be discerned when comparing the spectra for different salts at equal concentration of 1 M.”* (*J. Chem. Phys.* 141, 214502 (2014), ref. 22). The present work provides a clear rationale for these ion-specific trends.

Based on the reviewer’s above criticism, we realized that, for the sake of brevity in our original submission, we did not put our study into the right perspective. In the revised manuscript we have added this information to a new paragraph, where we emphasize the novelty of our results and the relation to earlier studies:

Page 2:

“Both, the dynamics of water in the solvation shell of ions and the motion of ions itself, go along with a change of the macroscopic dipole moment of the sample and can thus be probed using spectroscopy experiments.^{10,22–24} Here microwave and Terahertz spectroscopies have been extensively used. At field frequencies ranging from 100 MHz to ~100 GHz, hydration of ions has been intensively studied by detecting the dynamics of water and also the rotational dynamics of long-lived ionic aggregates (i.e. ion-pairs) have been elucidated in detail.^{24–26} At frequencies ranging from ~2 THz up to 20 THz, at which hydrogen-bonding vibrations, librations, and also ions contribute, the cooperativity and spatial extent of ion hydration, as well as ion-pairing, have been investigated in great detail.^{10,23,27,28} At intermediate frequencies (100 GHz – 2 THz) computational studies predict the very weak contribution of ionic currents to peak.²² Experimentally, this intermediate frequency range is, however, challenging to study and experiments on electrolytes are scarce.²⁹ As such, the potential of using this spectral information to understand ion dynamics has so far been not exploited.^{22,29”}

2. The experimental data (as shown in Figure 3) do not support the strong claims being made by the authors. The data would require an

improved analysis and much more data points in order to come to unambiguous conclusions.

In the following I want to address this point in more detail:

First I want to point out that the density of the solution is changing when increasing the concentration. The experimental data should be density corrected before any fit is done. I cannot find any statement on this in the paper.

We respectfully disagree with the reviewer in this point: Polarization, as defined by the complex permittivity, corresponds to polarization per volume unit. As such, it can be directly related to e.g. dipoles or ions per volume unit (i.e. molar concentrations). Thus, correction of the experimental spectra for density would lead to an incorrectly biased analysis.

The reviewer possibly refers to mixture studies using the Terahertz absorption coefficients as observable, where such a correction is indeed essential. This is however not the case for our work using permittivities as observable.

A summary of the experimental data is given in Figure 3a): Relaxation amplitude, S_{fast} , of the fast relaxation mode as a function of salt concentration, c_{salt} is displayed. They show linear fits to the data. For a concentration of 0, i.e. bulk water, a value of ca. 2.2 was fixed. What does this offset correspond to? The overall variation in S_{fast} (from 1.5 to 4.0) is rather small compared to this offset value.

We thank the reviewer for these insightful comments. The offset value corresponds to the spectral contributions due to water. Unfortunately, it is experimentally impossible to discriminate between ionic contributions and contributions due to water, as all contributions add up to the measured permittivity. Thus, we experimentally model all dynamics (being it due to water or ions) by a higher frequency Debye mode and contributions of water give rise to the offset.

We have explained this in more detail in the revised manuscript on page 4:

“To capture the faster dynamics at ~ 0.3 THz we use a Debye-type mode³² (see methods section for details). We note that both, motion of ions and motion of water molecules, contribute to the spectra at these frequencies.²² As such, we use the Debye mode as a means to quantify the contribution of all fast dynamics. The variation of the parameters of the Debye mode with concentration thus reflects the salt-induced changes to these dynamics.”

And on page 8:

“In contrast to the simulations, which can disentangle the contributions of ions from the contributions of water, we experimentally monitor all fast dynamics due to water and ions by studying the amplitude of the fitted Debye mode. As the motions of both water and ions contribute to the spectral intensity at ~ 0.3 THz (Fig. 1c), only the variation of the intensity with

salt content allows for drawing conclusions on ion-induced dynamics, while the absolute values of S_{fast} (and also τ_{fast}) also contain information on the dynamics of water."

Now, I want to take a closer look at the data:

A linear scaling with salt concentration is found for LiCl, KCl, KI, and NaSO₄ (only 2 points).

For GdmCl (3 points), MgSO₄ (3 points), KSCN (2 points only !) the displayed fit to a linear function does not make any sense. The points are far off the fitted line, they have large error bars, and do not show a linear trend.

We apologize for potential confusion. The linear fits in Fig. 3a were not intended to imply any physical meaning, but were just meant as a mere guide to the eye. To avoid this potential confusion, we have removed the linear fits from Fig. 3a and only show an overall linear fit to the data to discuss the correlation with concentration in the revised Fig. 3a.

In the previous studies cited above it was shown that the response is linear for ions like KCl, but becomes nonlinear in case of ion hydration. A nonlinear concentration dependence can be attributed directly to ion pairing.

The previous work by the Havenith group focuses on the spectra at frequencies above 2 THz, where hydrogen-bond stretching vibrations and water librational modes of water dominate. Since the frequency region in this paper differs from that in the previous work, the previous discussion cannot be directly applied to the present work.

Nonetheless, we agree with the reviewer that ion-pairing may lead to deviations from linearity also for the dynamics of our work, ion-pairing is only one of various possible factors that can reduce both conductivity and S_{fast} : ion-pairing signifies itself as a pronounced peak in the RDFs and results, as such, in a steeper potential of mean force. This steeper potential of mean force reduces the spatial extent of thermally accessible ion-excursions from the ion equilibrium position. As mentioned above, ion-pairing also trivially reduces the sample's conductivity.

However, the key finding of our study is the vastly different slopes for different salts of the fluctuation amplitude when plotting vs. concentration, while the slopes become more similar when plotting the fluctuation amplitude vs. conductivity (i.e., a correlation between the slopes of $S_{fast}(c)$ and $\kappa(c)$). We show that the correlation of long-ranged and short-ranged friction and varying distribution of ions, including ion-pairing as a limiting case, gives rise to this correlation. Thus, our results provide a general framework for fast ionic dynamics.

We have added this notion to the revised manuscript on page 10:

“This correlation can be understood by noting that microscopic and macroscopic friction are related and that the potential energy landscape of the ions in the cage determines the ions’ thermally accessible excursions, which are also related to the ion escaping its cage to allow for ion transport. This includes the frequently inferred concept of ion-pairing^{23,50}: electroneutral ion-pairs do not contribute to conductivity and the ions reside in very steep potentials that restrict the amplitude of the ionic motions.”

The authors state: “Remarkably, for a wide range of salts (KCl, LiCl, GdmCl, KI, KSCN, Na₂SO₄, MgSO₄) the experimentally determined amplitudes collapse onto a single curve: Starting from a value of ~2.1 for the fast dynamics of water at ~0 S m⁻¹ (neat water), the amplitude increases to 3.5 for electrolytes with a d.c. conductivity of ~25 S m⁻¹. Even for the strong acid HCl, where the charge transport mechanism is very different (Grothuss type transport with charge transport being decoupled from mass transport⁴⁶), the correlation between the conductivity and the fast relaxation strength holds.” Fig 3 c is not supporting this strong statement. Instead, it shows that the conductivity scales linearly with concentration for only a few ions (LiCl, KCl, and KI), which also do not show ion pairing. However, this result is not unexpected, the conductivity is expected to scale with concentration.

It can be seen that for the same value of $S_{fast} = 2.5$, conductivity values between ca. 3.5 and 12 can be found. Thus the statement in the paper as given above is not supported by the data, and these data do not collapse into a simple linear relationship!

We agree with the reviewer that we were somewhat overexcited by the finding and that the statement is too strong. Furthermore, we agree with the reviewer that linear scaling for conductivity with concentrations for dilute electrolytes is not surprising. However, our key finding is that the slopes of $S_{fast}(c)$ and $\kappa(c)$ are related, leading to the scaling as indicated in Fig 3c.

To emphasize the scaling, we discuss in the revised manuscript the overall correlation instead of the individual linear fits and focus on Pearson’s correlation coefficients for the $S_{fast}(c)$ and $S_{fast}(\kappa)$. Such analysis shows that S_{fast} is indeed only moderately correlated with the concentration ($r=0.64$), while correlation with the conductivity is strong ($r=0.82$). To further demonstrate that the reported scaling also holds for other electrolytes that hardly form ion-pairs, we have also included new experiments on MgCl₂, NaCl, CsCl in the revised manuscript. We have added this notion to the discussion on page 8 of the revised manuscript:

“As can be seen in Fig. 3a, the values for S_{fast} vary widely for different mono- and bivalent salts. However, salt concentration does not exclusively determine the magnitude of S_{fast} when comparing all studied salts (as one may expect from asymptotic electrolyte theory that treats

ions as point-like charges): The data in Fig. 3a are scattered, and we find a Pearson's correlation coefficient $r = 0.64$ for $S_{fast}(c_{salt})$."

and

"Remarkably, in contrast to the moderate correlation of S_{fast} with c_{salt} , the values of S_{fast} show a very strong correlation with the electrolyte conductivity (Pearson's correlation coefficient $r=0.82$, Fig. 3c) for a wide range of mono- and bivalent salts ($CsCl$, KCl , $NaCl$, $LiCl$, $GdmCl$, KI , $KSCN$, Na_2SO_4 , $MgSO_4$, $MgCl_2$): Starting from a value of ~ 2 for the fast dynamics at $\sim 0 S m^{-1}$ (neat water), the amplitude increases to ~ 3.5 for electrolytes with a d.c. conductivity of $\sim 25 S m^{-1}$."

We also had a closer look at the scatter of the data. While part of the scatter certainly results from the fact that we extract a small contribution of ions from a spectrum that is dominated by the response of water, we also could identify that data at $< 1 GHz$ can bias the fitted fast dynamics. At $< 1 GHz$ the spectra are sensitive to slight variation in the calibration with conductive silver paint, which becomes critical for samples with high conductivity (see Ref. 41 for details). Therefore, we restrict analysis to spectra at $\nu > 0.96 GHz$ in the revised manuscript. This restriction mainly affected the results based on fits using linear deviations (Fig. S23), but also resulted in slightly different values in Fig. 3.

Instead I think the data can be summarized alternatively in the following way: Higher S_{fast} values are found for all salts which contain K and which do not form ion pairs. Lower values are found for ion pair forming salts.

We thank the reviewer for his thoughts on the results. Based on the reviewer's comments we have performed experiments on $CsCl$, $NaCl$, and $MgCl_2$. These results together with our data demonstrate that also salts with cations other than K^+ can give rise to S_{fast} of > 2.5 ($LiCl$ and $CsCl$). These data have been included in the revised manuscript.

We agree that ion-pairing is one mechanism that can lower both S_{fast} and the conductivity. Ion-pairing is reflected in a very marked peak in the RDFs, which gives rise to a very steep potential and thus reduces the spectral contributions of the ions to the detected dynamics. Electroneutral ion-pairs, of course, do not contribute the macroscopic conductivity. Our results, however, show that also friction and the overall distribution of ions affect the spectral dynamics, with ion-pairing being an extreme case for the distribution of ions. Thus, we believe that our scaling is more general than just reflecting the degree of ion-pairing.

To emphasize that both friction and ion-distribution give rise to the observed ionic dynamics, we discuss the simulated ionic contributions of the charge scaled simulations quantitatively on page 7 of the revised manuscript:

“Quantitatively, we find the height of the simulated $I_{||}$ peaks (Fig. 2a) to vary by +50% (-25%), upon decreasing (increasing) the charge by 30%, while the diffusivities of the ions vary by +30% (-30%) (see Table S1, SI). In light of the oscillator model described above, the higher sensitivity of $I_{||}$ to the ionic charge, as compared to the diffusivity, shows that both the altered friction and the altered ionic distribution give rise to the changes in $I_{||}$ upon charge scaling.”

and added the notion that ion-pairing is one possible contribution to the observed scaling on page 10:

“This correlation can be understood by noting that microscopic and macroscopic friction are related and that the potential energy landscape of the ions in the cage determines the ions’ thermally accessible excursions, which are also related to the ion escaping its cage to allow for ion transport. This includes the frequently inferred concept of ion-pairing^{23,50}: electroneutral ion-pairs do not contribute to conductivity and the ions reside in very steep potentials that restrict the amplitude of the ionic motions.”

The authors state that “The relaxation time is rather similar for all studied salts, except for sulfate salts, which are prone to the formation of long-lived ion-pairs”. The same result was reported before in the review in *Angewandte Chemie* “The linewidths for all these bands reflect the damping of the vibrational modes of the ion-water complexes. Remarkably, we found that all free ions seem to be subject to identical damping, which means that they are all connected to the same thermal bath”

We disagree with the reviewer here. The review in *Angewandte* reports on the one hand on different dynamics at different frequencies. Even if these dynamic were related, the relaxation time τ_{fast} determines the ‘peak position’ of the Debye mode and not the bandwidth. Within the harmonic oscillator model, the damping coefficient determines the linewidth. In fact, the reported scaling with conductivity indicates that the damping is very different for different salts: the scaling is based on approximation of the damping from the molar conductivity of the ions, which is very different for different ions.

2. The authors fit the data using a damped harmonic oscillator, using the model of Schmidt et al. reference 38 for the frequency range between 50 and 300 cm^{-1} .

However, especially in the low frequency regime, a refined model should be used to fit the overdamped modes, following the Approach described in the book:

A. Nitzan, *Chemical Dynamics in Condensed Phases: Relaxation, Transfer, Reactions in Condensed Molecular Systems*, Oxford University Press, Oxford 2006.

We apologize for any ambiguity in our original submission. We fit the data using a Debye function and not using a harmonic oscillator. As already mentioned above, various dynamics contribute the spectral contributions in the frequency range 0.3-1 THz, which are experimentally impossible to discriminate. We only use the damped harmonic oscillator to illustrate the origin of the ionic contributions as obtained from the MD simulations. As we use the oscillator only for illustration purposes, we believe that any refined models will not provide additional information.

To emphasize that we use the oscillator only for illustration purposes, we have added the following notion on page 7 of the revised manuscript:

“Note that while the harmonic oscillator model does not capture all details of the simulated I_{II} spectra, it serves to illustrate the underlying molecular-level dynamics.”

Especially the case of HCl, where charge transport occurs via a Grotthuss mechanism, shows that the underlying model is much too simplified.

We thank the reviewer for this comment. Triggered by the comment, we have performed additional experiments at higher HCl concentrations. These experiments show that the reviewer is correct: due to the different conduction mechanism, the scaling between S_{fast} and κ breaks down, as the reviewer points out. Therefore we have moved the results for HCl to the SI and added the following note to the main text on page 8:

“We note that even for the strong acid HCl the correlation holds at low concentrations, while it breaks down at higher concentrations (see Fig. S20, SI), which can be related to the very different charge transport mechanism for the proton (Grothuss type transport with charge transport being decoupled from mass transport⁴⁹).”

4. In Figure 4, they display the ionic contributions of the dielectric loss spectrum of a simplified electrolyte showing that the spectra show contributions up to the third shell. Previously, using ab initio simulations, Schienbein et al have showed that the theoretical THz difference spectra of aqueous salt solutions can be deciphered in terms of only a handful of dipolar auto- and

crosscorrelations, including the second solvation shell, see J. Phys. Chem. Lett. 2017, 8, 2373–2380. This very detailed analysis showed that “Dramatic intensity cancellations due to large positive and negative contributions are found to effectively shift intensity maxima.”. For Cl⁻ the largest contribution below 50 cm⁻¹ arises from $\Delta C_{\text{Ion}}(\omega)$, which stems from the autocorrelation of the ion and its cross-correlations with water molecules not only in the first, but also in the second solvation shell. However, other negative contributions would become relevant for Br⁻.

This pioneering work by Marx *et al.* focused on higher frequency Terahertz modes, as opposed to the dynamics of the present study. Also the cross-correlation in J. Phys. Chem. Lett. 2017, 8, 2373–2380 refers to cross-correlation between the motion of ions and the motion of water. Such ion-water cross-correlations are dominated by short-ranged charge – dipole interactions.

The data in Figure 4 are indicative of correlated ionic motion, for which interaction is dominated by longer-ranged charge-charge interactions. Thus, correlations extend naturally over longer ranges (as in Debye-Hückel theory). Thus, the spatial extend of the correlations in our manuscript naturally extend over longer length scales as compared to the cross-correlations reported by Marx *et al.*

We note that we also observe negative contributions from ion-water cross-correlations (see I_{IW} in Figure 1c), similar to the reports by Marx *et al.*. We have added this notion to the revised manuscript on page 5:

“The correlation between the dynamics of water and the ions (I_{WI} in Fig. 1c) has a negative sign and ‘counters’ the ionic polarization, similar to what has been found at higher frequencies by Marx and coworkers.¹⁰”

Minor point: The last sentence “points towards the possibility of tuning the macroscopic conductivity in e.g. ionic liquids or battery electrolytes by molecular engineering” does not make sense, since it is not related to any aspect of the paper

We agree with the reviewer and revised this statement accordingly. The revised final sentence on page 10 now reads:

“The scaling reported here provides a rationale for understanding, and possibly engineering, the macroscopic conductivity in electrolytes, e.g., ionic liquids or battery electrolytes.”

Reviewer 2

This is a study of the molecular dynamics in the GHz-THz region of aqueous electrolyte solutions by dielectric relaxation spectroscopy and MD simulations. Currently, aqueous solutions of high concentrated electrolytes are getting more attention for application of battery, in particular. In this study, the authors have experimentally found a mild correlation between the electrical conductivity and the amplitude of the fast band at ~ 0.3 THz. MD simulations have pointed out that the THz fluctuations observed in the experiments are governed by the correlated motions of ions and water with the correlation extending up to the 3rd coordination shell which is longer than the Debye length. I think that the findings are interesting and helpful for the deeper understanding of aqueous electrolyte solutions. The manuscript is well-written overall. Though I have some (relatively minor) comments shown in below, I recommend this manuscript for publication in Nature Communications after they are addressed appropriately.

We thank the reviewer for carefully reading our manuscript and for his very positive assessment of our work.

(1) Relation between the electrical conductivity and the amplitude of the 0.3 THz band. I think that this new finding is very interesting. On the other hand, I do not understand it well. My concern is why the "amplitude" (not "time" or "frequency") of the band is related to the electrical conductivity? The amplitude of the band in the dielectric relaxation is coming from the concentration of the signal origin and the strength of the transition dipole moment. In the case of a single solute, the latter effect should be minor unless ions make some aggregations. However, if the solute changes like this study, the latter effect should be large. It is rather straight forward to understand that the concentration of solute is related to the electrical conductivity, but I do not understand how the strength of the transition dipole moment can influence the electrical conductivity. Adding some comments on this in the text would be helpful to understand the relation.

We thank the reviewer for his comment and we agree that the underlying molecular-level details were not very clear in our original submission.

The harmonic oscillator can be used to illustrate the scaling: The ions fluctuate around their equilibrium positions. With the ions at their equilibrium position, the net dipole moment of the sample is 0. A higher polarization due to ionic motion, as observed by higher spectral contributions to the dielectric spectra, can have two possible origins:

(i) the ions reside in shallower potentials and therefore thermally accessible displacements span larger distances (i.e. larger transition dipole). As such, the integral of the amplitude of

the oscillator (or the peak maximum) scales with $1/\sqrt{k}$. We argue that shallower potentials also go along with a lower activation barrier to escape the potential and this activation barrier determines conductivity (see also Fig. S11).

(ii) the ions experience lower friction, which results in reduced damping. This reduced damping results in higher polarization at ~ 0.3 THz and thus in an increased peak height (or peak amplitude) for such ion dynamics.

To better explain the possible mechanisms that affect the observed ionic dynamics, we discuss the oscillator model in more details in the revised manuscript on page 7:

“Note that while the harmonic oscillator model does not capture all details of the simulated $I_{||}$ spectra, it serves to illustrate the underlying molecular-level dynamics. Within this model, the peak maximum (or similar, the peak integral) of the oscillator inversely scales with damping, as higher friction attenuates the ions’ motion and thus reduces the ionic polarization according to $\epsilon_{HO}''(\omega_0) \sim c_{salt}/\omega_0\gamma$, see eq S13, SI. The oscillator amplitude also scales with the inverse fluctuation frequency, ω_0 , since steeper (shallower) potentials and/or heavier (lighter) ions narrow (widen) the spatial extent of thermally accessible excursions of the ions out from their equilibrium. We conclude that the $I_{||}$ spectral contributions sensitively report on the ionic distribution and dissipative ion effects (as quantified by the friction coefficient) in solution.”

Based on our simulations of the charge-scaled ions we can, in fact, demonstrate that both mechanisms described above are at play, as the variation of the simulated ionic fluctuations cannot be solely explained by the changed friction (judged by the diffusivities). We have added this discussion accordingly on page 7 of the revised manuscript:

“Quantitatively, we find the height of the simulated $I_{||}$ peaks (Fig. 2a) to vary by +50% (-25%), upon decreasing (increasing) the charge by 30%, while the diffusivities of the ions vary by +30% (-30%) (see Table S1, SI). In light of the oscillator model described above, the higher sensitivity of $I_{||}$ to the ionic charge, as compared to the diffusivity, shows that both the altered friction and the altered ionic distribution give rise to the changes in $I_{||}$ upon charge scaling.”

In the experiments, we are most sensitive to variation in the spectral amplitudes. This can be understood by the fact that at the relevant frequencies different dynamics contribute, including ionic motion but also motion of water. Experimentally it is impossible to isolate the ionic dynamics, and the band position of the fitted Debye mode will depend on the relative weight of the ionic contributions relative to the water contributions. However, given the broad nature of the fitted Debye peak, any additional spectral amplitude due to the presence of the ions will result in a higher amplitude (increased polarization) of this ‘composite’ mode. As such, while other parameters certainly vary, the experimental quantity that is most sensitive to ion-induced changes is the amplitude of the fast relaxation.

We explain this in more detail in the revised manuscript on page 4:

“We note that both, motion of ions and motion of water molecules, contribute to the spectra at these frequencies.²² As such, we use the Debye mode as a means to quantify the contribution of all fast dynamics. The variation of the parameters of the Debye mode with concentration thus reflects the salt-induced changes to these dynamics.”

and on page 8

“In contrast to the simulations, which can disentangle the contributions of ions from the contributions of water, we experimentally monitor all fast dynamics due to water and ions by studying the amplitude of the fitted Debye mode. As the motions of both water and ions contribute to the spectral intensity at ~ 0.3 THz (Fig. 1c), only the variation of the intensity with salt content allows for drawing conclusions on ion-induced dynamics, while the absolute values of S_{fast} (and also τ_{fast}) also contain information on the dynamics of water.”

(2) Terminology of “relaxation”. In this manuscript, the authors use “fast relaxation” for the band at ~ 0.3 THz (for example, page 8), which is attributed the ions’ fluctuation. It is commonly used the word “relaxation” for an overdamped motion. But I feel that it might be not very appropriate for using “relaxation” for “fluctuation”, an “underdamped motion”, or “vibrational motion”. When it is used for a “vibrational motion”, it often means its dephasing process. If the authors assign the motion at ca. 0.3 THz as the ions’ fluctuation, I recommend to use other word, such as motion or dynamics.

We thank the reviewer for this comment and we agree that dynamics is a more appropriate term and we have changed the term wherever possible.

We, however, left the term relaxation, when discussing the parameters of the Debye mode as its corresponding parameters (relaxation time and relaxation strength) are the unambiguous and widely used terms for these parameters.

(3) Harmonic oscillator model. It is a bit surprising and not surprising as well that the harmonic oscillator model can work for aqueous KI solution. The authors simply consider the pair of K^+ and I^- . If I look at Figure 4, however, the situation is not like that simple even in 0.5 M: it seems to be necessary to consider the many body effect. In such case, is the reduced mass appropriate? Some comments for this in the text should be added.

We fully agree with the reviewer. The oscillator model is just a means to illustrate the underlying molecular mechanisms. It cannot represent the ionic motion in an electrolyte quantitatively and additional effects like hydrations shells or other coordination shells will

certainly affect the reduced mass. Also, the harmonic approximation will certainly lead to deviations. We have added a comment on possible deviations due to the reduced mass to the revised manuscript on page 6:

“The predicted maximum of the spectral response of the harmonic oscillator model is shifted by a factor of ~4 to higher frequencies as compared to the simulated $I_{||}$ response (Fig. 2a), which is attributable to the anharmonicity of the potential and ions beyond the first coordination shell, neglected in our harmonic approximation (Fig. 2c, see also Fig. S10, SI). Also an underestimation of the reduced mass, as hydration of ions and/or electrostatic interaction with other ions may effectively result in a higher reduced mass, could contribute to this difference.”

(4) Figure 3. It is hard to distinguish between KCl and MgSO₄. I recommend replacing with a distinguishable color for either one.

We fully agree, and we have changed the colors of this figure and used different symbols for different salts.

(5) Typo. Page 8 line 8: “3 coordination shell” should be “3rd coordination shell”.

We thank the reviewer for spotting this typo. The text now reads “*three coordination shells*”

Reviewers' comments:

Reviewer #1 (Remarks to the Author):

Macroscopic conductivity of aqueous electrolyte solutions scales with ultrafast microscopic ion motions

By: Vasileios Balos,^{1,†} Sho Imoto,¹ Roland R. Netz,² Mischa Bonn,¹ Douwe Jan Bonthuis,^{2,3,*} Yuki Nagata,^{1,*} and Johannes Hunger^{1,*}

While part of the remarks and comments have been taken into account, I still have major concerns as to whether the experimental results support the claims being made.

a) Compared to the previous versions, the authors have now left out the data below 1 GHz. This implies that parts of the spectrum are not included in the analysis.

Based on the SI that, for some of the salts, the fitted band is partially outside of the investigated frequency range.

I'm afraid I must say that I lost confidence in the stability of the fit when I compared the data from the previous version to this most recent version.

Some of the values differ considerably, e.g. for GdmCl at a concentration of 0.5 mol L⁻¹, S_{fast} was originally ca. 1.5; it is now ca. 2.0. In the case of KSCN at a concentration of 1.5 mol L⁻¹, S_{fast} was deduced to be ca. 3.5, however, now a value of ca. 2.6 is shown. In contrast, for a concentration of 2.5 mol L⁻¹ the value of KSCN is almost unchanged compared to the previous version. These make no sense to me and I am unable to confirm the reproducibility of the results.

For $S_{\text{fast}}=2.5$ the conductivity ranges now from ca. 4 to 12.5.

b) The proposed correlation between ϵ'' and κ is based on the following equations, outlined in the SI:

$$\epsilon'' \sim 2 c_{\text{salt}} q^2 / \epsilon_0 1/(\omega_0 \gamma)$$

$$\kappa \sim c_{\text{salt}} q^2 / (kD T) (D^+ + D^-)$$

The proportionality of each of these to c^{salt} is trivial. Thus, the more important question is whether $\epsilon''/c_{\text{salt}}$ and κ/c_{salt} are correlated.

Looking into the details of the new figure, I speculate that the correlation (0.82) of S_{fast} with the conductivity is still mainly based on the data for KI, since only these data include high conductivities (besides one KCN value). Omitting the KI data and seeing how much this affects the correlation could easily test this. The other data are scattered between 2.0 and 3.0 with considerable error bars, which do not even account for systematic errors such as neglecting the low frequency part.

The linear concentration dependence of the conductivity for KI is not an unexpected result.

Moreover, I would suggest plotting the dependence $\epsilon''/c_{\text{salt}}$ versus κ/c_{salt} instead of Figure 3, which would be a better test. The Pearson's Correlation coefficient should be deduced from these numbers.

In their answer, the authors write: "The linear fits in Fig. 3a were not intended to imply any physical meaning, but were just meant as a mere guide to the eye. To avoid this potential confusion, we have removed the linear fits from Fig. 3a and only show an overall linear fit to the data to discuss the correlation with concentration in the revised Fig. 3a."

I am surprised about this change compared to the previous paper. In their model (see above) S_{fast} should be correlated with the concentration. If their model holds, the different slopes would reflect the distinct $1/(\omega_0 \gamma)$ values, but the linear correlation for each salt should still hold.

According to their model the amplitude should vary linearly with the concentration, however, the slope should be different. Also, a slope is expected for other models – as long as ion pairing does not dominate. However, in none of the cases is the same slope used for all salts.

Thus, I find an overall fit misleading and it cannot be used to prove or disprove any model.

General remark: thermodynamically, the molality, i.e. the number of anions/cations per water molecule is the more relevant parameter.

In summary, I feel unable to approve the validity of the statistical analysis in its present form.

Reviewer #2 (Remarks to the Author):

This is a revised manuscript that I reviewed previously. I have found that the authors have addressed all my comments appropriately. I therefore believe that this manuscript is worth publishing in Nature Communications.

While part of the remarks and comments have been taken into account, I still have major concerns as to whether the experimental results support the claims being made.

a) Compared to the previous versions, the authors have now left out the data below 1 GHz. This implies that parts of the spectrum are not included in the analysis.

Based on the SI that, for some of the salts, the fitted band is partially outside of the investigated frequency range.

I'm afraid I must say that I lost confidence in the stability of the fit when I compared the data from the previous version to this most recent version.

Some of the values differ considerably, e.g. for GdmCl at a concentration of 0.5 mol L⁻¹, S_{fast} was originally ca. 1.5; it is now ca. 2.0. In the case of KSCN at a concentration of 1.5 mol L⁻¹, S_{fast} was deduced to be ca. 3.5, however, now a value of ca. 2.6 is shown. In contrast, for a concentration of 2.5 mol L⁻¹ the value of KSCN is almost unchanged compared to the previous version. These make no sense to me and I am unable to confirm the reproducibility of the results.

For S_{fast}=2.5 the conductivity ranges now from ca. 4 to 12.5.

While indeed some of the bands are at the lower edge of our frequency range, which is limited due to the intrinsic Ohmic loss of the electrolyte solutions, the fast mode is covered by the experimental data for all salts.

The reviewer negatively comments the changed values of individual data points for the fast mode as a result of a slightly restricted frequency range at lower frequencies used in the analysis of the revised manuscript, yet completely ignores the error bars: For 0.5 mol/L GdmCl the value in Figure 3 increased from 1.51±0.43 to 2.01±0.14 and the error bars overlap. For 1.5 mol/L KSCN the value decreased from 3.47±0.17 to 2.86±0.33, which is slightly beyond the error range. Out of the 48 data points, reanalysed in Figure 3 and Figure S23, the restriction at lower frequencies lead only for 6 values to changes that exceed the error bars, mostly by < 0.1 and in two cases by 0.1 and 0.2, which can be – as pointed out by the reviewer – ascribed to neglecting systematic errors. As such, we emphatically disagree with the concerns about the reproducibility.

The proposed correlation between ϵ'' and κ is based on the following equations, outlined in the SI:

$$\epsilon'' \sim 2 c_{\text{salt}} q^2 / \epsilon_0 1/(\omega_0 \gamma)$$

$$\kappa \sim c_{\text{salt}} q^2 / (kD T) (D^+ + D^-)$$

The proportionality of each of these to c^{salt} is trivial. Thus, the more important question is whether $\epsilon''/c_{\text{salt}}$ and κ/c_{salt} are correlated.

Looking into the details of the new figure, I speculate that the correlation (0.82) of S_{fast} with the conductivity is still mainly based on the data for KI, since only these data include high conductivities (besides one KCN value). Omitting the KI data and seeing how much this affects the correlation could easily test this. The other data are scattered between 2.0 and 3.0 with considerable

error bars, which do not even account for systematic errors such as neglecting the low frequency part.

The linear concentration dependence of the conductivity for KI is not an unexpected result. Moreover, I would suggest plotting the dependence $\varepsilon''/c_{\text{salt}}$ versus κ/c_{salt} instead of Figure 3, which would be a better test. The Pearson's Correlation coefficient should be deduced from these numbers.

The reviewer claims that our correlation arises from the data for KI: We disagree with this, as the trend is unchanged even if we ignore the data for KI; omitting the data for KI change the correlation coefficient for $S_{\text{fast}}(c)$ from 0.64 to 0.67, while it also changes the correlation for $S_{\text{fast}}(\kappa)$ from 0.86 to 0.82. As a result, the difference between the correlation of $S_{\text{fast}}(\kappa)$ and $S_{\text{fast}}(c)$ is still apparent, even when data for KI are omitted.

The suggestion of the reviewer to compare the slopes of the conductivity and the fast fluctuation amplitude versus concentration is a valid point. However, we intentionally did not show the correlation of the slopes as this is a correlation of fit parameters obtained from fits to inherently noisy data (we extract the small amplitude of ionic motion with an amplitude less than 2 from a spectrum that is dominated by water's response with an amplitude of >70). Nevertheless, the suggested analysis shows also a positive correlation between the slopes of $S_{\text{fast}}(c)$ and $\kappa(c)$ with a Pearson's correlation coefficient of $r=0.42$. We note here that omission of the data for KCl and Na_2SO_4 , which are based on only two samples, would make the correlation substantially better:

Correlation between the slopes of $S_{\text{fast}}(c)$ and $\kappa(c)$. Grey symbols correspond to the values for KCl and Na_2SO_4 , which are determined from only two samples with different salt concentrations.

Although this analysis clearly also demonstrates the correlation with conductivity, the above considerations led us to consider the overall correlation of $S_{\text{fast}}(c)$ and $S_{\text{fast}}(\kappa)$ as we show in Figure 3 of our manuscript, which shows a better correlation and a $\sim 50\%$ reduction of the sum of the square

residuals for $S_{\text{fast}}(\kappa)$ as compared to $S_{\text{fast}}(c)$, making it the most appropriate way to demonstrate the correlation of the inherently noisy data.

In their answer, the authors write: "The linear fits in Fig. 3a were not intended to imply any physical meaning, but were just meant as a mere guide to the eye. To avoid this potential confusion, we have removed the linear fits from Fig. 3a and only show an overall linear fit to the data to discuss the correlation with concentration in the revised Fig. 3a."

I am surprised about this change compared to the previous paper. In their model (see above) S_{fast} should be correlated with the concentration. If their model holds, the different slopes would reflect the distinct $1/(\omega_0 \gamma)$ values, but the linear correlation for each salt should still hold.

According to their model the amplitude should vary linearly with the concentration, however, the slope should be different. Also, a slope is expected for other models - as long as ion pairing does not dominate. However, in none of the cases is the same slope used for all salts. Thus, I find an overall fit misleading and it cannot be used to prove or disprove any model.

About the reviewer's comments on the meaning of $S_{\text{fast}}(c)$: Although we agree that $S_{\text{fast}}(c)$ should have a physical meaning, as detailed above, the inherently small amplitude of the ionic motions makes it challenging to accurately determine $S_{\text{fast}}(c)$. As such, we refrain from ascribing a physical meaning due to the uncertainties in determining these slopes.

General remark: thermodynamically, the molality, i.e. the number of anions/cations per water molecule is the more relevant parameter.

We do not disagree with this statement, yet we need to point out that in a spectroscopic study the number of ions per volume unit determines the measured polarization, and thus the spectroscopic data cannot be directly analyzed using thermodynamically relevant quantities.

REVIEWERS' COMMENTS:

Reviewer #3 (Remarks to the Author):

The revised version of this manuscript is well-written and many additional details are given in the supplementary material.

All concerns made by the previous reviewer are addressed adequately. The authors have demonstrated that their analysis is sound.

My minor issue concerns the computation of the ionic contribution to the dielectric spectrum. Due to the periodic boundary conditions $M_I(t)$ should contain large jumps of $\pm 1e \cdot 36$ Angstroem for each ion reinserted at the other side of the simulation box. Usually this fact prohibits the use of this quantity in equilibrium correlation functions. How did the authors cope with this problem? However, this is a minor technical detail which does not question the data analysis.

Overall, the manuscript in its present form is definitely worth publishing in Nature Communications! The authors have done a marvellous job and the current work is very interesting for the community. I do recommend the publication without any further delay.

Reviewer 3:

The revised version of this manuscript is well-written and many additional details are given in the supplementary material. All concerns made by the previous reviewer are addressed adequately. The authors have demonstrated that their analysis is sound.

My minor issue concerns the computation of the ionic contribution to the dielectric spectrum. Due to the periodic boundary conditions $M_I(t)$ should contain large jumps of $\pm 1e \cdot 36$ Angstrom for each ion reinserted at the other side of the simulation box. Usually this fact prohibits the use of this quantity in equilibrium correlation functions. How did the authors cope with this problem? However, this is a minor technical detail which does not question the data analysis.

We thank the reviewer for this comment and apologize for the omission of this detail. To avoid the above-described polarization jumps, we computed the itinerant polarization as described by Sprik et al. (Supplementary Reference 11 of the revised manuscript), for which the polarization is obtained from the positions of the atoms and water molecules that are not relocated in the simulation box.

We have added this notion to Supplementary Note 2 of the revised manuscript, which reads: "*Here, we use the itinerant polarization, for which r_i represents the positions of ions and water molecules that are not relocated back to the primary simulation cell.*"¹¹

Overall, the manuscript in its present form is definitely worth publishing in Nature Communications! The authors have done a marvellous job and the current work is very interesting for the community. I do recommend the publication without any further delay.

We are delighted by the reviewer's positive assessment and the very encouraging comments.